# The Implication of *Vibrio* Bacteria in the Winter Mortalities of the Critically Endangered *Pinna nobilis*

**DOI:** 10.3390/microorganisms9050922

**Published:** 2021-04-26

**Authors:** Athanasios Lattos, Konstantina Bitchava, Ioannis A. Giantsis, John A. Theodorou, Costas Batargias, Basile Michaelidis

**Affiliations:** 1Laboratory of Animal Physiology, Department of Zoology, Faculty of Science, School of Biology, Aristotle University of Thessaloniki, 541 24 Thessaloniki, Greece; lattosad@bio.auth.gr; 2Laboratory of Fish, Shellfish & Crustacean Diseases, Veterinary Research Institute of Thessaloniki ELGO-DEMETER, 570 01 Thessaloniki, Greece; bitchava@vri.gr; 3Department of Animal Science, Faculty of Agricultural Sciences, University of Western Macedonia, 531 00 Florina, Greece; 4Department of Animal Production Fisheries & Aquaculture, University of Patras, 232 00 Mesolonghi, Greece; jtheo@upatras.gr (J.A.T.); cbatargias@upatras.gr (C.B.)

**Keywords:** *Pinna nobilis*, mass mortality, *Vibrio* spp., *Vibrio splendidus*, *Vibrio mediterranei*, *Haplosporidium pinnae*, *Mycobacterium* sp., antibiotics

## Abstract

*Pinna nobilis* populations, constituting the largest bivalve mollusk endemic to the Mediterranean, is characterized as critically endangered, threatened by extinction. Among the various factors proposed as etiological agents are the *Haplosporidium pinnae* and *Mycobacterium* sp. parasites. Nevertheless, devastation of the fan mussel populations is still far from clear. The current work is undertaken under a broader study aiming to evaluate the health status of *Pinna nobilis* population in Aegean Sea, after the mass mortalities that occurred in 2019. A significant objective was also (a) the investigation of the etiological agents of small-scale winter mortalities in the remaining populations after the devastating results of *Haplosporidium pinnae* and *Mycobacterium* sp. infections, as well as (b) the examination of the susceptibility of the identified bacterial strains in antibiotics for future laboratory experiments. Microbiological assays were used in order to detect the presence of potential bacterial pathogens in moribund animals in combination with molecular tools for their identification. Our results provide evidence that *Vibrio* bacterial species are directly implicated in the winter mortalities, particularly in cases where the haplosporidian parasite was absent. Additionally, this is the first report of *Vibrio mediterranei* and *V. splendidus* hosted by any bivalve on the Greek coastline.

## 1. Introduction

Marine habitats constitute natural hosts for a plethora of microorganism communities, which play key roles in fundamental functions among each ecosystem [1]. Microbes are often engaged in a mutualistic symbiosis with many inhabitants in marine environments, assisting them in mechanisms such as immune functions, physiological responses in various factors, and nutrient uptake [2,3,4]. Apart from the beneficial properties sometimes offered to the hosts, microbes can also act pathogenically under certain conditions and may become opportunistic, leading to disease pathogenesis [5]. Stressful conditions as a result of climate change can have a negative impact on the physiological responses of hosts, as well as implicating with lower immune responses [6,7]. At the same time, an increase in the average temperature can favor the colonization of bacterial genera such as *Vibrio* spp., inducing their virulence factors [8]. However, parasitism can expose the host species to secondary infections, which, together with the main causative agent, can increase the pathogenicity at the expense of the host [9,10].

Mortalities in *Pinna nobilis* populations persist in threatening the species towards extinction. This phenomenon started on the Spanish coastline in 2016 (Alicante, Spain), reaching mortalities at a ratio of 100%, infecting *P. nobilis* populations in all life stages, and was directly associated with the presence of a new emerging Haplosporidan parasite [11,12]. Morphological and molecular characterization were performed one year later alongside phylogenetic analysis, which confirmed the taxonomic status of the newly emerged parasite in the order Haplosporida [13]. Mortalities are still spreading in the Mediterranean Sea populations of *P. nobilis*, with the addition of a new potential candidate causative agent of mortalities. *Mycobacterium* species were detected alongside *H. pinnae* in the Tyrrhenian Sea (Mediterranean) in moribund individuals of *P. nobilis*, causing strongly inflammatory lesions, and were proposed as a new potential threat for the populations of fan mussels [14]. Meanwhile, the detection of *H. pinnae* persisted in mortalities in the Ionian and Aegean Sea, having a great negative impact on populations of fan mussels and causing the change of the ranking of the species in the Mediterranean Sea to Critically Endangered according to the IUCN red list [15,16,17,18,19,20,21]. Both aforementioned pathogens have been detected in *P. nobilis* populations suffering from mortalities on Croatian coastlines, thus invading the last safe shelter of the species from the mortalities that occurred in the rest of Mediterranean Sea [22,23]. The only parasite-free area is the Sea of Marmara, which was recently referred as unaffected from the epidemic [24]. Apart from the presence of the pathogens mentioned before, new potential pathogens were detected for the fan mussel in a survey conducted in Italy (Campania, Tuscany, Sardinia, and Apulia) and Spain (Catalunya). The presence of *Vibrio* spp. and *Perkinsus* sp. alongside mycobacteria and *H. pinnae* have been suggested to represent a harmful combination in disease pathogenesis against the fan mussel populations [25]. In a rescue project for *P. nobilis* conducted in Catalunya, a new pathogen emerged in individuals free from the parasites mentioned above. *V. mediterranei* was detected in dead individuals when the temperature rose between April and May (17–19 °C) in stable animals, and was later demonstrated to be infectious with the presence of specific virulence genes [26,27]. In the context of exploration in the *P. nobilis* microbiome, identification of eight different bacterial taxa was performed and the results concluded the presence of *V. splendidus* clade bacteria [28]. The multifactorial attributed mortality scenario was strengthened by a molecular survey conducted in Italy, which demonstrated that *H. pinnae* is not species-specific, but had occurred in marine bivalves since 2014, thus challenging the finding that *H. pinnae* is the unique etiological agent of the mortalities of *P. nobilis* in Mediterranean Sea [29]. In a recent study, investigating the biochemical performance of the species, specimens collected before and after the mortality events were compared, and the results provided evidence that *Mycobacterium* sp. existed in the species long time ago before mortalities occurred [30]. The theory of a pathogenic cluster of agents, which causes mortalities, was also supported in a study conducted using, for the first time, the powerful tool of next-generation sequencing of the 16s-rRNA gene in *P. nobilis* individuals. The microbiome of individuals from three different populations, previously described to be infected from *H. pinnae* and *Mycobacterium* sp. [17], were analyzed, and revealed the presence of 14 different bacterial OTUs, among which were bacterial genera such as *Vibrio* sp., *Mycoplasma* spp., *Pseudoalteromonas* spp., *Mycobacterium* sp., *Aliivibrio* spp., *Photobacterium* spp., and *Psychrilyobacter* spp. [31]. The aforementioned genera can be symbiotic or pathogenic genera for bivalves, supporting the complexity of mass mortality events. However, even before the onset of mortalities with pathogenic microorganisms as causative agents, the *P. nobilis* population along the Mediterranean Sea faced human pressure, which resulted in a serious decline in the population in the last few decades. Illegal fishing for seafood or decorative purposes, anchoring, bycatches, and habitat destruction have pushed the populations towards their lower survival limits [32,33]. 

In the context of continuous mortalities in Greece, the current study was conducted in order to assess the status of the remaining *P. nobilis* populations in Greek territory. After the collapse of fan mussel populations in Thermaikos gulf, Thessaloniki [19], the remaining populations with the highest density, which were chosen for investigation, were the populations in Kalloni gulf, Lesvos Island and Maliakos gulf, Fthiotis (Figure 1). The main objectives of this study were (a) the monitoring of the two remaining populations in Greece regarding haplosporidian and mycobacteria parasites during the winter months; (b) identification of other harmful bacteria such as *Vibrio* spp. that have been detected by *P. nobilis* microbiome analysis; (c) investigation of small-scale mortalities in the aforementioned gulfs; and, furthermore, (d) to determine the most effective antibiotics that could possess antimicrobial activity against the newly identified *Vibrio* species. Taken altogether, the study aims to enlighten and contribute to the determination of the etiological agents of the fan mussel *P. nobilis* mortalities.

## 2. Materials and Methods

### 2.1. Sampling

In order to evaluate the status of the remaining *P. nobilis* in the last refugia in Greece, samplings of *P. nobilis* specimens were performed in 2 different marine areas in the Aegean Sea, i.e., Kalloni gulf, Lesvos island and Maliakos gulf, Fthiotis during February 2020 and April 2020 (Figure 1). Thirty moribund samples were collected in 2020. Ten samples originated from Kalloni gulf, Lesvos island, divided in two sampling efforts with equal samples each, and the other twenty were from Maliakos gulf, Fthiotis, divided into ten samples in each sampling effort.

All samplings were carried out in compliance with the terms of a particular license received from the Greek Ministry of Environment and Energy (code: MEE//GDDDP89926/1117). *P. nobilis* specimens were dissected immediately after collection in aseptic conditions. Each tissue was divided in small parts and each part was placed in sterilized 1.5ml tubes and stored in liquid nitrogen for further analyses. Additionally, a small part of the digestive gland was kept and fixed for histological process.

### 2.2. Histological Procedure

Histopathological examination of the samples was performed exactly as described in Lattos et al. [17] in an effort to evaluate the pathological condition of the tissues, as well as the potential differences in infections caused by different microorganisms.

### 2.3. Microbiological Methods

#### 2.3.1. Culture Media 

Tryptic soy agar (TSA, Oxoid) was used as the nonselective media, and thiosulfate citrate bile salts sucrose (TCBS, Oxoid) for the isolation of *Vibrio* strains. They were both prepared according to the manufacturer’s instructions, with the addition of 2% NaCl (Oxoid). Additionally, tryptic soy broth (TSB, Oxoid) with 2% NaCl was used for the cryopreservation of the strains at −80 °C in a final dilution of 15% glycerol.

#### 2.3.2. Bacterial Isolation

One gr of the different pen shell tissues (mantle, gill, digestive gland, and muscle) were homogenized with 9 mL of sterile PBS buffer [26]. 100 μL of homogenate was used to inoculate on TSA and TCBS agar supplemented with 2% NaCl. The plates were incubated for 24–48 h at 25 °C and observed for bacterial growth. For isolation, single colonies were picked and stroked onto new plates (Figure 2). Liquid cultures were incubated overnight at 25 °C. All isolates that were used for further identification and analysis were the ones found dominant in the first step of inoculation on TSA and TCBS.

#### 2.3.3. Antibiogram

The susceptibility of *Vibrio* species to various antimicrobials was determined through the disk diffusion method, according to the guidelines of the Clinical and Laboratory Standards Institute [34]. The antibiograms in all strains isolated and identified were prepared using 15 antibiotic disks: florfenicol (FFC, 30 μg), erythromycin (E, 15 μg), cephalothin (KF, 30 μg), cefotaxime (CTX, 30 μg), ampicillin (AMP, 10 μg), amoxicillin/clavulonate (AMC, 20 μg and 10 μg, respectively), kanamycin (K, 30 μg), neomycin (N, 30 μg), gentamycin (GM, 10 μg), streptomycin (S, 10 μg), trimethoprim sulfamethoxazole (SXT, 1.25 μg and 23.75 μg, respectively), ciprofloxacin (CIP, 5 μg), flumequine (UB, 30 μg), norfloxacin (NOR, 10 μg), and tetracycline (TE, 30 μg). 

The bacterial strains were inoculated into Mueller–Hinton broth (Oxoid) containing 2% NaCl and incubated overnight at 25 °C. Afterwards, each suspension was adjusted to the turbidity of a 0.5 McFarland standard and spread onto Mueller–Hinton agar (Oxoid) plates containing 2% NaCl. The antibiotic disks were placed on the plates that were then incubated for 24 h at 25 °C. The diameter of the inhibition zone around each disk was measured and recorded. The results were classified as resistant (R), intermediately resistant (I), or susceptible (S) according to the CLSI guidelines [34].

### 2.4. Molecular Identification of Pathogens

Pure cultures of bacteria stored at −80 °C were incubated onto TSB with 2% NaCl for 24–48 h for the extraction of DNA. DNA extraction was conducted using the DNAEasy Blood and Tissue kit (QIAGEN, Germany) according to the manufacturer’s instructions. The quality and quantity of the isolated DNA was evaluated spectrophotometrically in a NanoDrop (Shimadzu, Japan). Approximately 50 ng of extracted DNA were subjected in a 20 μL volume PCR using 10 μL of the FastGene Taq 2x Ready Mix (NIPPON Genetics, Europe) and 0.6 pmol of each one of the primers 27f-CM and 1492r modified by Frank et al. [35] that amplify a c. 1200 part of the 16Sr RNA gene in bacteria. The following PCR conditions were applied: initially, a denaturation step of 2 min at 94 °C, followed by 36 cycles of 40 s at 94 °C, 45 s at 50 °C, and 1 min at 72 °C, and, in the end, by a final extension step of 5 min at 72 °C. After electrophoresis of the PCR products in agarose gel stained with ethidium bromide, they were purified using the NucleoSpin Gel and PCR Clean-up kit (Macherey-Nagel, Germany) and were sequenced bi-directionally in an ABI-PRISM 3130xl genetic analyzer. Sequences were read, aligned, and phylogenetically resolved using the MEGA 7.0 software [36]. Using the same software, two phylogenetic dendrograms were constructed implementing the maximum likelihood methodology and applying 1000 bootstrap iterations to evaluate the genetic relationships with closely related haplotypes available in GenBank. Furthermore, to explore the disease pathogenesis of the *Vibrio* identified bacteria, the presence of virulence factors was examined, i.e., *vsm* and outer membrane protein (*ompU*) genes in *V. splendidus* strains and ompU and rtx toxin genes in *V. mediterranei* strains, using the FastGene Taq 2x Ready Mix with the abovementioned volumes, and primers and conditions as previously described [27,37].

In addition, the potential presence of *Mycobacterium* sp. and *H. pinnae* was examined in all tissue samples before bacterial isolation, using the exact same PCR conditions, primers, and procedures as described in Lattos et al. [17].

## 3. Results

### 3.1. Molecular Identification of Pathogenic Bacteria

All examined samples were found positive to *Mycobacterium* sp., whereas only 3 out of the 17 were positive to *H. pinnae* infection (Table 1). In regard to bacteria cultures, alignment was based on a 950 bp long partial sequence of the 16r RNA gene that defined ten different haplotypes, all belonging to various species of the genus *Vibrio* (Table 1, Figure 3). In particular, two phylogenetic trees were constructed, the first of which included all new *Vibrio* haplotypes in comparison to congeneric haplotypes available in the GenBank database (Figure 3a). The haplotype GR139 was identical with *V. gigantis* strains, whereas the haplotypes GR252 and GR146 were very closely related with *V. crassostreae* clades. The strains GR131, GR142, and GR147 were phylogenetically very closely related with—and grouped among—*V. splendidus* haplotypes. Since *V. mediterranei* was not previously reported in the Aegean Sea and has been characterized as an invasive emerging pathogen, a second tree was also constructed comparing the haplotype GR246 with only *V. mediterranei* sequences (Figure 3b). Sequence similarity of this haplotype was always more than 98.5%, and up to 100% with *V. mediterranei* sequences obtained from the GenBank database, and was therefore grouped within *V. mediterranei* haplotypes. All novel sequences were deposited in the Genbank database under the accession numbers (MW715023–MW715032).

Eventually, to explore the perspective of disease pathogenesis, the presence of virulence factors was investigated in *V. splendidus* and *V. mediterranei* infected samples. Five out of the nine *V. splendidus* strains were positive for both *vsm* and *ompU* virulence factor genes, three only for *ompU,* and one was negative for both virulence factor genes, whereas the *V. mediterranei* strain was positive for both *ompU* and rtx virulence factor genes (Table 2).

### 3.2. Antibiograms

All strains were susceptible to the sulfonamide, phenicol, fluoroquinolones, tetracycline, and third generation cephalosporin tested, while they exhibited variable results considering the first-generation cephalosporin, penicillins, macrolide, and aminoglycosides tested. The results are shown in Table 3. There were only 2 strains out of the 17 showing resistance to the first generation cephalosporin or cephalothin, 9 out of the 17 resistant to ampicillin, and 5 to amoxicillin/clavulonate. From the aminoglycoside group, six resistances to neomycin, three to streptomycin and two to kanamycin were recorded. Finally, 2 out of 17 were resistant to erythromycin. All tested strains were susceptible to the sulfonamide, the phenicol, the fluoroquinolones, and the tetracycline. Seven of the strains tested were sensitive to all antibiotics used, while 3 out of the 17 strains tested were the ones with the most resistances encountered. These strains were identified as *V. mediterranei* and *V. crassostreae*, all of which originated from Maliakos gulf, Fthiotis and Kalloni gulf, Lesvos island, respectively.

### 3.3. Histological Evaluation of the Inflammatory Responses

Despite the absense of *H. pinnae* (Figure 4B), specimens infected with *V. splendidus* showed histopathological lesions to a similar degree as specimens hosting the haplosporidian parasite in the digestive gland (Figure 4A). In the same way, heavy inflammatory responses were observed in all specimens, whether or not the *H. pinnae* was present. Diffuse type inflammation was observed in all specimens.

## 4. Discussion

### 4.1. Molecular Characterization of Vibrio Species

The strain GR246, found in Maliakos gulf, was grouped within *V. mediterranei* clade strains (Figure 3b), providing evidence for its identification as *V. mediterranei*. *V. shilloni*, which was also very closely phylogenetically related with this group, is a coral pathogen [38,39], whereas *V. mediterranei* is a pathogen of bivalves that was recently found infecting *P. nobilis* as well. Keeping this in mind, we can propose that the derived *Vibrio* haplotype in the specimens originating from Maliakos gulf belongs to the species *V. mediterranei.* To the best of our knowledge, this is the first report of *V. mediterranei* in the Aegean Sea isolated from *P. nobilis*. 

At least three different *Vibrio* haplotypes were clustered within the *V. splendidus* group. Specifically, haplotype GR139 was identical to several *V. splendidus*, *V. gigantis*, and *V. crassostreae* strains, while haplotypes GR252 and GR146 represent newly described sequences, more closely phylogenetically related to *V. chagasi* than to *V. splendidus* and *V. crassostreae*. Although, in this case, the phylogeny is too complicated to extract safe conclusions regarding the molecular taxonomy of this strain; based on the life history and geographic distribution of each species, we can postulate that it is probably *V. splendidus*. Notably, this is also the first report of *V. splendidus* hosting *P. nobilis* in the Greek seas. 

Furthermore, the isolate GR247 detected with *P. nobilis* specimens from Kalloni gulf, Lesvos island was genetically close within the same clade, and particularly in the same branch as *V. toranzoniae*. This species was first described in 2013, hosting healthy reared individuals of the clams *Venerupis philippinarum* and *Venerupis decussata* originating from Galicia, Spain [40]. 

Finally, two different isolates, phylogenetically very closely related, were grouped within the *V. harveyi* group. Both isolates represent novel haplotypes. Despite the fact that, similarly to the case of *V. splendidus* clade, no safe conclusions can be considered regarding their molecular identification; it should be pointed that these strains were genetically distinct from all the remaining ones.

### 4.2. Microbiological Characterisation of Vibrio Species and Susceptibility in Antibiotics

As far as the antibiograms that were conducted in this study are concerned, we did not record important resistances in the strains tested. Antimicrobial resistance among halophilic *Vibrio* species isolated from oysters, mussels, and clams was studied in Canada for 6 years and reported that only 4.9% of the strains used were sensitive to all drugs tested, while in our study, fewer resistances were reported [41]. According to the same authors, similar antimicrobial resistance patterns to ours were detected reporting ampicillin, cephalothin, erythromycin, kanamycin, and streptomycin to be the antibiotics with the higher recorded resistances. Additionally, it has been reported that more than 90% of the *Vibrio* isolates tested showed resistance to streptomycin, whereas many strains (50%) were resistant to penicillin, carbenicillin, ampicillin, cephalothin, and kanamycin [42]. Τhese resistances are attributed to the environmental pressure posed by various human activities [41]. 

The antibiotic profile of the strains was conducted not only for a possible selection of a drug to be used in order to treat an infection, but also represents a marker of the dominant spatial environmental conditions, considering the antibiotic resistance genes that are exchanged between the most dominant bacteria, such as bacteria belonging to the *Vibrio* species. 

It is interesting to note that, for the Ampicillin, there is a significant difference between the two populations. However, this is at the limits of Bonferroni correction. Additionally, there is a valid statistical difference between the two sites regarding the resistance to antibiotics in general, with the bacteria detected in fan mussels originating from Kalloni being more resistant to antibiotics (*p*: 0.000, G-test: 185.5, df = 3). This is of particular importance on account of the continuous efforts to detect healthy fan mussel populations to be used in regeneration attempts.

### 4.3. Potential Impacts on Mortalities

Coastal and estuarine environments constitute marine mollusk habitat areas, which, due to their filter-feeding behavior, accumulate a large amount of bacterial microbiota [43]. In the marine environment, bivalve mollusks constitute habitats for the Vibrionaceae family and for other bacterial species [8]. To date, 142 species have been reported within the Vibrionaceae family, classified in seven genera, with the genus *Vibrio* representing the most variable one [44,45]. However, members of the genus *Vibrio* have been described and characterized as causative agents for many mass mortality episodes worldwide, affecting all the life stages of bivalves [43]. Bacteria have formed symbiotic relationships with their hosts, benefiting them in digestion, in nutrient absorption, in defense mechanisms against infectious pathogens, and in the formation of reproduction strategies. Conversely, the host provides a stable environment with a constant supply of nutrients for the symbiotic bacteria [46,47,48]. Some of them are also vital for natural systems, including the carbon cycle and osmoregulation [49]. There is a wide diversity of pathogenic bacteria belonging to the genus *Vibrio* associated with mass mortalities in marine bivalve mollusks [50]. Among the pathogenic *Vibrio spp.*, vibrios belonging to the splendidus clade have been described many times as causative agents of pathogenicities in marine organisms, and, in particular, to bivalves. *V. splendidus* has been reported worldwide hosted by the Pacific oyster, *Crassostrea gigas*, causing mortalities of up to 100% of the infected population in several cases [10,51,52,53,54]. However, *V. splendidus* clade bacteria have a wide range of bivalve mollusk hosts, such as scallops, mussels, and clams, causing similar harmful effects as in oysters [55,56,57,58,59,60]. Vibriosis in bivalve mollusks, caused by bacteria belonging to the splendidus clade, is associated with histopathological lesions, such as the disorganization of muscle fibers and strong inflammatory responses, as well as the general depression of the physiology [56,61]. Another pathogenic agent, belonging to Vibrionaceae family, causing mortalities in a wide variety of aquatic animals is *V. harveyi*, although it has been shown to belong to the microbiome of most of the infected hosts [62]. *V. harveyi* strains have been detected in many hosts such as crabs, teleosts, crustaceans, and bivalve mollusks, causing immune and physiological depression and mortalities [63,64,65,66,67,68]. *V. mediterranei* is considered to be an emerging pathogen, being abundant in aquatic environments, infecting hosts such as corals and seaweeds and causing mortalities synergistically with other pathogens [69,70]. This *Vibrio* species was from plankton, sediments, and seawater samples from two coastal areas in south Valencia, Spain [71].

Concerning the mortalities in fan mussel populations in the Mediterranean Sea, two potential pathogens were firstly identified as the etiological agents causing the mortalities. These pathogens were *H. pinnae* and *Mycobacterium* sp.; many other potential pathogens were also detected in samples originating from areas where mortalities occurred, making it more difficult to fully understand the phenomenon of these extended declines in the populations of the species. Nevertheless, recent studies, both in the field and in laboratory conditions, involve the presence of *Vibrio* species among the pathogens of fan mussel populations. Particularly *Vibrio* spp. of the Splendidus clade and *V. mediterranei* were detected in a field study and in laboratory conditions, respectively, hosting the fan mussel [26,28]. The current study demonstrated the existence of *Vibrio* bacterial strains in moribund samples of *P. nobilis,* originated from Maliakos gulf, Fthiotis and Kalloni gulf, Lesvos island, the last two surviving populations across the Greek coastline. Analysis of the low-scale winter mortalities demonstrated differential patterns of pathogen abundance among the specimens collected from the mortality events that took place in 2020 in the Greek coastal areas. *H. pinnae* abundance was limited, present only in three samples from the total of thirty collected, assuming that its spread is probably restricted by the lower temperatures. All specimens infected by the haplosporidian parasite originated from Maliakos gulf, Fthiotis. On the other hand, mycobacteriosis continued to exist in all fan mussel specimens in a 100% ratio in both examined areas. 

Concerning the histopathological results, no correlation can be observed with the presence of *Vibrio* spp., and this can be attributed to the complexity of the mortalities in fan mussels. *P. nobilis* populations face a multi-pathogen situation alongside the direct and indirect human pressures. Prevalence of mycobacteriosis is 100% in all populations investigated across the Greek coastline during the last three years. With the presence of this chronic disease, *P. nobilis* populations are affected by a chronic stress resulting in a general vulnerability to diseases.

According to our results, low-temperature mortalities may be attributed to many potential pathogens, acting as causative agents for the reduction of the natural populations, even in the winter. *V. splendidus* strains, which were abundant in the majority of the examined specimens, have been reported to exhibit their highest virulence at low temperatures, especially considering the presence of virulence genes, while their pathogenicity is reduced at increased temperatures [58,72]. Virulence genes upregulate the transmission and the reproduction rate of the pathogen [27]. The presence of virulence genes, particularly in *V. splendidus* and *V. mediterranei*, support the scenario of pathogenicity of vibrios in the Aegean Sea, as well as their contribution in winter mortalities. Our results therefore provide evidence that the devastation of the population of the fan mussel is far from clear, suggesting that many microorganisms may be implicated in this phenomenon. More epidemiological studies are needed, particularly from unaffected populations, to enlighten these phenomena. The Sea of Marmara was recently determined as such an area where healthy P. nobilis individuals, free of parasites, were detected [24]. Presumably, the outflow from the Dardanelles towards the Aegean Sea, forming a physical gene flow barrier from the Mediterranean in the direction of the Sea of Marmara [73], still prevents the expansion of the P. nobilis pathogens in this direction. Considering the harmful effects of climate change in combination with human pressure, despite the legal restrictions (illegal fishing, destruction of habitats), an action plan is considered as an emergency measure to prevent the extinction of the species, taking into account unaffected populations.

## Figures and Tables

**Figure 1 microorganisms-09-00922-f001:**
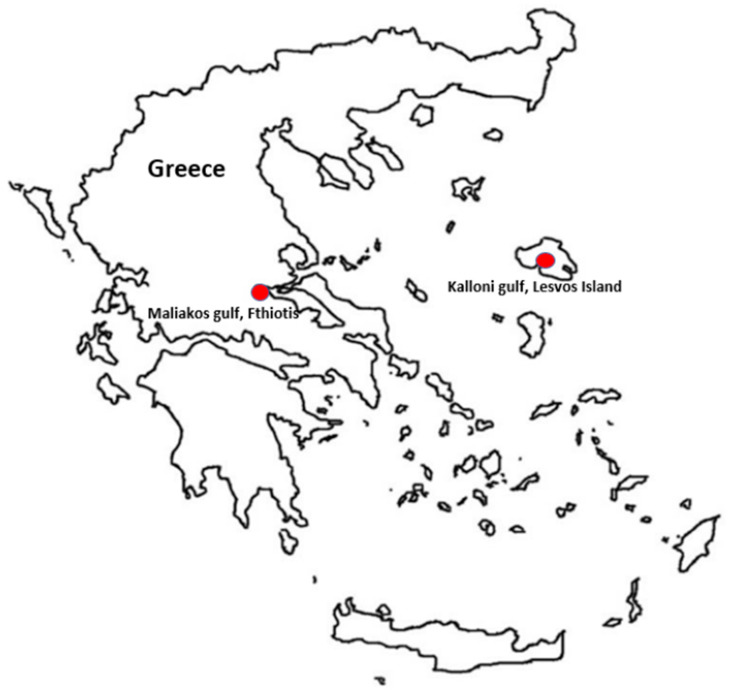
Sampling sites of P. nobilis specimens from Kalloni gulf, Lesvos Island (39.095818, 26.149199) and Maliakos gulf, Fthiotis (38.805781, 22.613020), within the Aegean Sea.

**Figure 2 microorganisms-09-00922-f002:**
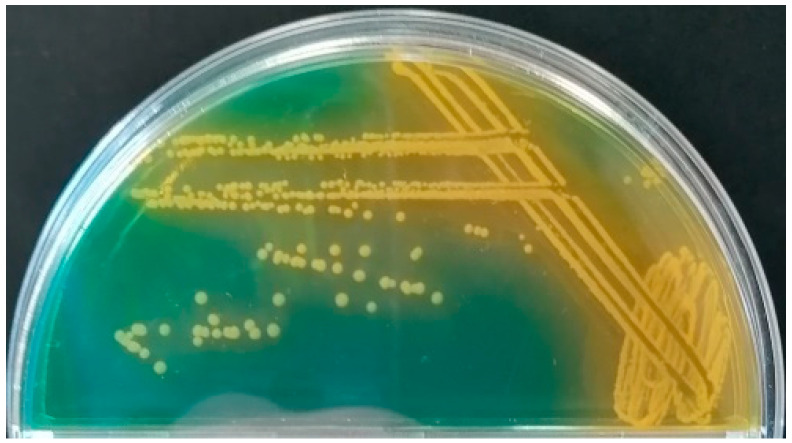
Subculture of a single colony on TCBS with 2% NaCl.

**Figure 3 microorganisms-09-00922-f003:**
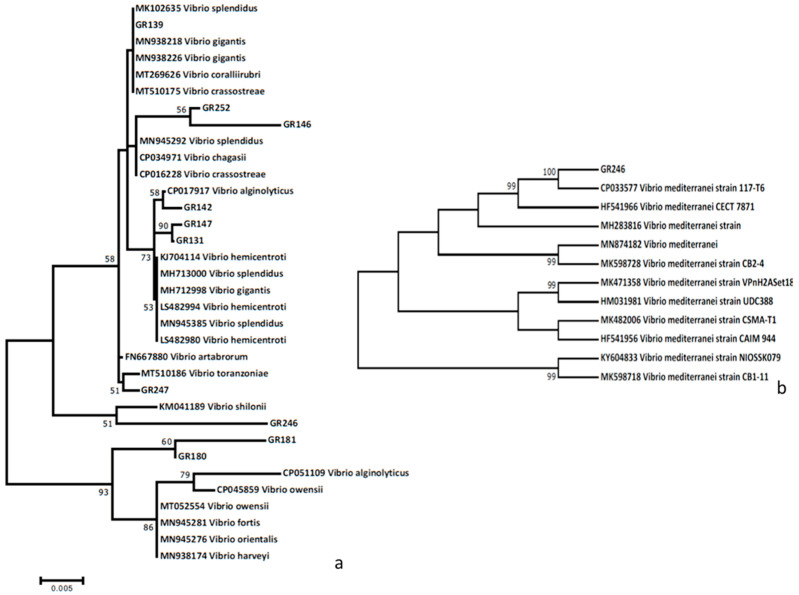
Phylogenetic relationship of the newly described *Vibrio* haplotypes in comparison with the most closely related ones, obtained from the GenBank database. (**a**) Maximum likelihood dendrogram including all newly described *Vibrio* haplotypes in comparison with the most closely related ones. (**b**) Maximum likelihood dendrogram including only *V. mediterranei* haplotypes. Haplotype names as in Table 1**.**

**Figure 4 microorganisms-09-00922-f004:**
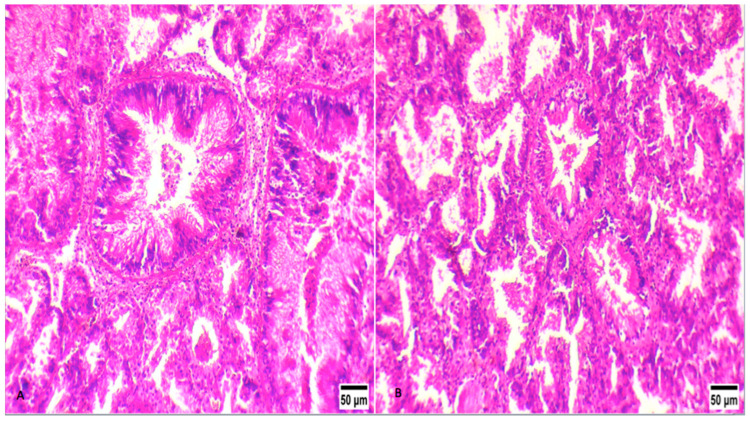
Histological display of the digestive gland of two different *P. nobilis* specimens. (**A**) Digestive gland of fan mussel infected with *Mycobacterium* sp., *H. pinnae*, and *V. splendidus*. (**B**) Digestive gland of fan mussel infected only with *Mycobacterium* sp. and *V. splendidus*. Both exhibit heavy inflammation responses and heavy lesions in the connective tissue of the digestive gland. A degenerative process was also indicated in the epithelium of the digestive tubules of both specimens, regardless of the presence of *H. pinnae*.

**Table 1 microorganisms-09-00922-t001:** Sampling data of the animals analyzed and the results of microbiological analysis in animal tissues. M, P, G, and GA are the abbreviations for mantle tissue, posterior adductor muscle, gill tissue, and digestive gland, respectively.

Sample ID	Code	Tissue	Sampling Date	Sampling Site	Geographical Coordinates (N)	Geographical Coordinates (E)	Sampling Depth	Temperature of Sampling Site (°C)	Habitat Type	*H. pinnae* PCR	*Mycobacterium* sp. PCR	*Vibrio spp*. PCR	Most Probable Taxonomy	Genbank A.N.
1	GR129	M	25 February 2020	Maliakos gulf	38.905781	22.613020	4–8 m	14.2	Soft substrate with *C. nodosa* meadows	+	+	+	*V. splendidus*	^1^ MW715032
2	GR131	P	25 February 2020	Maliakos gulf	38.905781	22.613020	4–8 m	14.2	Soft substrate with *C. nodosa* meadows		+	+	*V. splendidus*	MW715032
5	GR132	G	25 February2020	Maliakos gulf	38.905781	22.613020	4–8 m	14.2	Soft substrate with *C. nodosa* meadows		+	+	*V. splendidus*	^2^ MW715031
4	GR133	G	25 February 2020	Maliakos gulf	38.905781	22.613020	4–8 m	14.2	Soft substrate with *C. nodosa* meadows		+	+	*V. splendidus*	^3^ MW715028
6	GR134	M, G, GA	25 February2020	Maliakos gulf	38.905781	22.613020	4–8 m	14.2	Soft substrate with *C. nodosa* meadows		+	+	*V. splendidus*	^2^ MW715031
7	GR139	M, G	25 February2020	Maliakos gulf	38.905781	22.613020	4-8 m	14.2	Soft substrate with *C. nodosa* meadows		+	+	*V. gigantis*	MW715031
3	GR142	P	25 February 2020	Maliakos gulf	38.905781	22.613020	4–8 m	14.2	Soft substrate with *C. nodosa* meadows	+	+	+	*V. alginolyticus*	MW715030
4	GR144	G, GA, P	17 March 2020	Kalloni gulf	39.095818	26.149199	4–8 m	15.5	Soft Substrate		+	+	*V. splendidus*	^2^ MW715031
2	GR145	M, P, GA, G	17 March 2020	Kalloni gulf	39.095818	26.149199	4–8 m	15.5	Soft Substrate		+	+	*V. splendidus*	^2^ MW715031
3	GR146	M, P, GA, G	17 March 2020	Kalloni gulf	39.095818	26.149199	4–8 m	15.5	Soft Substrate		+	+	*V. gigantis*	MW715029
5	GR147	GA, P	17 March 2020	Kalloni gulf	39.095818	26.149199	4–8 m	15.5	Soft Substrate		+	+	*V. splendidus*	MW715028
1	GR180	M, GA	17 March 2020	Kalloni gulf	39.095818	26.149199	4–8 m	15.5	Soft Substrate		+	+	*V. owensii*	MW715027
1	GR181	M, G	17 March 2020	Kalloni gulf	39.095818	26.149199	4–8 m	15.5	Soft Substrate		+	+	*V. harveyi*	MW715026
9	GR245	GA, P	15 April 2020	Maliakos gulf	38.905781	22.613020	4–8 m	16.5	Soft substrate with *C. nodosa* meadows	+	+	+	*V. splendidus*	^3^ MW715028
8	GR246	GA	15 April 2020	Maliakos gulf	38.905781	22.613020	4–8 m	16.5	Soft substrate with *C. nodosa* meadows		+	+	*V. mediterranei*	MW715025
2	GR247	GA	5 May 2020	Kalloni gulf	39.095818	26.149199	4–8 m	16.1	Soft Substrate		+	+	*V. crassostreae*	MW715024
4	GR252	P	5 May 2020	Kalloni gulf	39.095818	26.149199	4–8 m	16.1	Soft Substrate		+	+	*V. crassostreae*	MW715023

^1^ Haplotype of the same sample as GR131, ^2^ haplotype of the same sample as GR145, ^3^ haplotype of the same sample as GR147.

**Table 2 microorganisms-09-00922-t002:** Presence of virulence factors in *V. splendidus* and *V. mediterranei* identified bacteria. Positive samples are indicated with “+” and negative samples are indicated with “-”.

Code	*Vibrio* Species	Virulence Genes			
		ompU	Vsm	ompU	rtx
GR129	*V. splendidus*	+	+		
GR131	*V. splendidus*	-	-		
GR132	*V. splendidus*	+	-		
GR133	*V. splendidus*	+	+		
GR134	*V. splendidus*	+	+		
GR144	*V. splendidus*	+	+		
GR145	*V. splendidus*	+	-		
GR147	*V. splendidus*	+	-		
GR245	*V. splendidus*	+	+		
GR246	*V. mediterranei*			+	+

**Table 3 microorganisms-09-00922-t003:** List of the strains and the antibiotics used for the antibiograms (KF: cephalothin, CTX: cefotaxime, AMP: ampicillin, AMC: amoxicillin/clavulonate, K: kanamycin, N: neomycin, GM: gentamycin, S: streptomycin, E: erythromycin, SXT: trimethoprim sulfamethoxazole, FFC: florfenicol, CIP: ciprofloxacin, UB: flumequine, NOR: norfloxacin, TE: tetracycline). The results are depicted by R: resistant and S: susceptible.

STRAIN	Cephalosporins		Penicillins		Aminoglycosides				Macrolide	Sulfonamide	Phenicol	Fluoroquinolones			Tetracycline
	KF	CTX	AMP	AMC	K	N	GM	S	E	SXT	FFC	CIP	UB	NOR	TE
144	S	S	R	S	S	S	S	S	S	S	S	S	S	S	S
245	S	S	S	S	S	S	S	S	S	S	S	S	S	S	S
131	S	S	S	S	S	S	S	S	S	S	S	S	S	S	S
132	S	S	S	S	S	S	S	S	S	S	S	S	S	S	S
139	S	S	R	S	S	R	S	S	S	S	S	S	S	S	S
129	S	S	S	S	S	S	S	S	S	S	S	S	S	S	S
142	S	S	S	S	S	S	S	S	S	S	S	S	S	S	S
146	S	S	R	S	S	R	S	S	S	S	S	S	S	S	S
181	S	S	R	R	S	S	S	S	S	S	S	S	S	S	S
180	S	S	R	R	S	S	S	S	S	S	S	S	S	S	S
133	S	S	S	S	S	R	S	S	S	S	S	S	S	S	S
134	S	S	S	S	S	S	S	S	S	S	S	S	S	S	S
145	S	S	R	S	S	S	S	S	S	S	S	S	S	S	S
147	S	S	S	S	S	S	S	S	S	S	S	S	S	S	S
246	S	S	R	R	R	R	S	R	S	S	S	S	S	S	S
247	R	S	R	R	S	R	S	R	R	S	S	S	S	S	S
252	R	S	R	R	S	R	S	R	R	S	S	S	S	S	S

## Data Availability

Data is contained within the article.

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
