# Peer review of "The Implication of Vibrio Bacteria in the Winter Mortalities of the Critically Endangered Pinna nobilis"

_microorganisms, 2021, doi:10.3390/microorganisms9050922_

Round 1
Reviewer 1 Report
The paper of Lattos et al deals with the MMs of Pinna nobilis and the involvement of bacterial disease, due to Vibrio sp. , to during winter mortality. This is the second paper written by the authors the first time approaching with NGS. The paper underline the potential role of Vibrio species, mainly V. splendidus, an important bacteria in the marine environment. The paper in my opinion is interesting including a new pathogen in the list in an area and need few amendments before to be published:
Check for the English. Some word looks weird in the way they’re used: example: page 2, line 88”declaration”. Even if it sounded like that, I think that the correct word they’re looking for is “finding”. Review the english please.
Introduction
Page 2, line 76:” In a rescue project for P. nobilis conducted in Italy…..” correct he sentence, the rescue project was in Catalunya.
Effect on the host
Because the authors don’t have healthy individuals included in the study, as it is happening in every work because of lack of animals, I think that:
- The authors should consider to check on virulence genes of the isolated colonies. V. splendidus strain contains a variable set of virulence genes, explaining their observed variable virulence for aquatic animals when tested independently. In V. splendidus for example these genes include potential toxins such as haemolysins, proteases, type VI secretion system, genes form siderophore transport and utilization and adhesins. This could open finally the perspective of talk of disease pathogenesis and not only of pathogen presence/absence.
- Pathogen related to lesions: have the authors found lesions in the animals connected to bacteria?
Have they thought to use a probe to detect the pathogen to the lesion? This could be interesting for to detect pathogen tropism and triggered lesions.
- Figure 3. I would advise to divide into 2 different trees for Vibrio, with one independent tree for mediterranei. The authors should detail better the identity in gene bank for this bacteria in the tree and describe that in the results.
Author Response
The paper of Lattos et al deals with the MMs of Pinna nobilis and the involvement of bacterial disease, due toVibrio sp. , to during winter mortality. This is the secondpaper written by the authors the first time approaching withNGS. The paper underline the potential role of Vibriospecies, mainly V. splendidus, an important bacteria in themarine environment. The paper in my opinion is interesting including a new pathogen in the list in an area and needfew amendments before to be published:
Response:We are grateful to the 1streviewer for the time and effort to revise our manuscript, as well as for his/her valuable comments, the vast majority of which has been taken into consideration in the revised manuscript. Below, a precise and point-by-point response to each comment is presented and we therefore believe that the amendments needed have been included.
Check for the English. Some word looks weird in the way they’re used: example: page 2, line 88”declaration”. Even if it sounded like that, I think that the correct word they’re looking for is “finding”. Review the english please.
Response: “Declaration” has been replaced by “finding”, as suggested by the reviewer. Also, several parts of the manuscript were checked for the English language, and accordingly, modifications in word that looked weird were carried out in cases needed.
Introduction
Page 2, line 76:” In a rescue project for P. nobilis conducted in Italy…..” correct he sentence, the rescueproject was in Catalunya.
Response: Corrected according to the comment of the reviewer.
Effect on the host
Because the authors don’t have healthy individuals included in the study, as it is happening in every work because of lack of animals, I think that:
- The authors should consider to check on virulence genes of the isolated colonies. V. splendidus strain contains a variable set of virulence genes, explaining their observed variable virulence for aquatic animals when tested independently. In V. splendidus for example these genes include potential toxins such as haemolysins, proteases, type VI secretion system, genes form siderophore transport and utilization and adhesins. This could open finally the perspective of talk of disease pathogenesis and not only of pathogen presence/absence.
Response: Based on the reviewer’s aforementioned comment, four (4) PCR reactions were additionally performed in order to check for virulence factors, i.e. vsm and outer membrane protein (ompU) genes in Vibrio splendidus strains and ompU and rtx toxin genes in Vibrio mediterranei. The corresponding parts in the Materials and methods, Results and Discussion, describing in detail the procedure followed and the obtained results were added or modified.
- Pathogen related to lesions: have the authors found lesions in the animals connected to bacteria? Have they thought to use a probe to detect the pathogen to the lesion? This could be interesting for to detect pathogen tropism and triggered lesions.
Response: No lesions were detected, correlated exclusively with Vibrio infection, probably due to the multifactorial infection of various microorganisms. Nonetheless, a new histopathological figure was added to evaluate and compare the status between specimens hosting Haplosporidium pinnae and specimens free of H. pinnae. Also, a small paragraph was added in the Discussion concerning these inferences.
- Figure 3. I would advise to divide into 2 different trees for Vibrio, with one independent tree for mediterranei. The authors should detail better the identity in gene bank for these bacteria in the tree and describe that in the results.
Response: Taking into consideration the reviewer’s comment, the phylogenetic analysis was reassessed, constructing two different phylogenetic trees for Vibrio, one of which was independent only for V. mediterranei. Also, the identity in gene bank for these trees was re-described in detail.
Reviewer 2 Report
In the article entitled "Is it vibriosis that kills the critically endangered Pinna nobilis? Identification, microbiological and phylogenetic analysis of multiple Vibrio species hosting the fan mussel from mortality events in the Aegean Sea (E. Mediterranean)" the authors presented the results of isolation and identification of Vibrio sp. from a P. nobilis coming from marine areas in the Aegean Sea, i.e. Kalloni gulf, Lesvos island and Maliakos Gulf, Fthiotis. The main aim of the research was to explain and identify the etiological factors causing P. nobilis mortality. To achieve this goal, the authors used standard microbiological methods to identify microorganisms and determine their drug susceptibility. The authors conclude from their research that Vibrio sp. may cause greater mortality of P. nobilis in winter, especially when they were not infected with the parasite Haplosporidian. These studies also showed the presence of Vibrio mediterranei and V. splendidus in these organisms on the Greek coast, for the first time. Research is well planned but the manuscript needs improvement. Comments and questions to the authors were given below:
- In my opinion, the title requires correction. It is too long and therefore does not clearly indicate what this article is about. It is also difficult for me to understand the first part of this title which is the question.
- the introduction is very extensive and contains a lot of irrelevant information that distracts from the main topic of the work. Especially it concerns the fragment on the increase in mortality in Pinna nobilis populations (page 2, lines 50-90)
- authors should check the concentration of antibiotics in the discs they provided in the "Material and methods" section 2.2.3.9 The concentration is in mg for sure?
- the description of the results from the drug susceptibility analysis should be more detailed in the part of the results and in the general and summary form in part of the discussion. Now it's the other way around
- the manuscript contains many typing and editing errors and needs to improve the English language
Author Response
In the article entitled "Is it vibriosis that kills the critically endangered Pinna nobilis? Identification, microbiological and phylogenetic analysis of multiple Vibrio species hosting the fan mussel from mortality events in the Aegean Sea (E. Mediterranean)" the authors presented the results of isolation and identification of Vibrio sp. from a P. nobiliscoming from marine areas in the Aegean Sea, i.e. Kalloni gulf, Lesvos island and Maliakos Gulf, Fthiotis. The main aim of the research was to explain and identify the etiological factors causing P. nobilis mortality. To achieve this goal, the authors used standard microbiological methods to identify microorganisms and determine their drug susceptibility. The authors conclude from their research that Vibrio sp. may cause greater mortality of P.nobilis in winter, especially when they were not infected with the parasite Haplosporidian. These studies also showed the presence of Vibrio mediterranei and V.splendidus in these organisms on the Greek coast, for the first time. Research is well planned but the manuscript needs improvement. Comments and questions to the authors were given below:
Response: We would like to thank the reviewer for his/her valuable and constructive comments. We believe that after the modifications conducted, based on both reviewers’ comments, the manuscript has been improved and now meets the reviewer’s requirements.
- In my opinion, the title requires correction. It is too long and therefore does not clearly indicate what this article is about. It is also difficult for me to understand the first part of this title which is the question.
Response: The title has been shortened, as suggested by the reviewer, clarifying better the concept of the study.
- the introduction is very extensive and contains a lot of irrelevant information that distracts from the main topic of the work. Especially it concerns the fragment on the increase in mortality in Pinna nobilis populations (page 2, lines 50-90)
Response: In accordance with the reviewer’s comment, a large part of the introduction has been deleted, avoiding non-necessary elements that may confuse the reader.
- authors should check the concentration of antibiotics in the discs they provided in the "Material and methods" section 2.2.3.9 The concentration is in mg for sure?
Response: We apologize for this mistake. Indeed, the correct concentration of antibiotics is μg and therefore ithas been corrected according to the comment of the reviewer.
- the description of the results from the drug susceptibility analysis should be more detailed in the part of the results and in the general and summary form in part of the discussion. Now it's the other way around
Response: A more detailed description of the results concerning the drug susceptibility analysis has been provided (please see 3.2 Antibiograms) reducing the corresponding part of the discussion, as proposed by the reviewer.
- the manuscript contains many typing and editing errors and needs to improve the English language
Response: Several parts of the manuscript were checked for the English language, and were accordingly modified and corrected. We now believe the quality of the written English language has been substantially improved